# Description of the Expansion of a Two-Layer Tube: An Analytic Plane-Strain Solution for Arbitrary Pressure-Independent Yield Criterion and Hardening Law

**Sergei Alexandrov** [1,2] , **Elena Lyamina** [3] **and Lihui Lang** [1,*]

[1]  School of Mechanical Engineering and Automation, Beihang University, Beijing 100191, China; sergei_alexandrov@spartak.ru
[2]  Faculty of Materials Science and Metallurgy Engineering, Federal State Autonomous Educational Institution of Higher Education, South Ural State University (National Research University), 454080 Chelyabinsk, Russia
[3]  Ishlinsky Institute for Problems in Mechanics RAS, 119526 Moscow, Russia; lyamina@inbox.ru
*  Correspondence: lang@buaa.edu.cn; Tel.: +86-10-82316821

**Abstract:** The main objective of the present paper is to provide a simple analytical solution for describing the expansion of a two-layer tube under plane-strain conditions for its subsequent use in the preliminary design of hydroforming processes. Each layer's constitutive equations are an arbitrary pressure-independent yield criterion, its associated plastic flow rule, and an arbitrary hardening law. The elastic portion of strain is neglected. The method of solution is based on two transformations of space variables. Firstly, a Lagrangian coordinate is introduced instead of the Eulerian radial coordinate. Then, the Lagrangian coordinate is replaced with the equivalent strain. The solution reduces to ordinary integrals that, in general, should be evaluated numerically. However, for two hardening laws of practical importance, these integrals are expressed in terms of special functions. Three geometric parameters for the initial configuration, a constitutive parameter, and two arbitrary functions classify the boundary value problem. Therefore, a detailed parametric analysis of the solution is not feasible. The illustrative example demonstrates the effect of the outer layer's thickness on the pressure applied to the inner radius of the tube.

**Keywords:** tube hydroforming; two-layer tube; rigid plasticity; arbitrary yield criterion; arbitrary hardening law; analytic solution

## 1. Introduction

Tube hydroforming is capable of replacing several traditional manufacturing processes [1]. The products of tube hydroforming processes are widely used in different sectors of the industry [2–6], including the production of micro-parts [7,8]. Several comprehensive reviews on hydroforming technologies are available [9–12], where the advantages and disadvantages of these technologies are discussed in detail.

An important direction of research is to design hydroforming processes. Several methods based on sophisticated numerical modeling have been proposed [1,5,13]. However, it is known from other branches of the mechanics of metal forming processes that simplified methods can be very useful for the preliminary design of metal forming processes. In particular, such methods can provide a reliable initial guess for more sophisticated methods. An example of such simplified methods is the theory of ideal flows [14–16]. The present paper provides a simple analytic solution for a two-layer tube hydroforming process. An advantage of this solution is that it is valid for any pressure-independent yield criterion and any hardening law. Therefore, the solution can be used for parametric analysis and preliminary design of the tube hydroforming process for a large class of materials.

Hydroforming of multi-layer materials is a widely used hydroforming process. The hydroforming technology for producing double-layer spherical vessels was introduced

in [17]. This paper includes both experimental and theoretical results. The latter are based on the elastic/plastic finite element method. Paper [18] has applied hydroforming for developing the discrete layer forming of a multi-layer tube. This paper uses an analytic method for finding an optimal loading path to prevent defects in the course of forming. This analytic method has been justified by experiments. The hydraulic bulging test for multi-layer sheets was proposed in [19]. Its theoretical treatment has been based on the finite element method. Various technological aspects of the hydroforming process of multi-layer sheets have been discussed in recent publications [20,21].

The present paper focuses on a new theoretical method for describing two-layer tube hydroforming under plane-strain conditions. It is assumed that each layer is rigid/plastic. No restriction is imposed on the isotropic pressure-independent yield criterion and hardening law. The general solution is analytic. A numerical treatment may be needed for evaluating ordinary integrals.

The success of the method proposed is based on the use of advantageous space variables. In particular, the original formulation of the boundary value problem in Eulerian coordinates is first transformed in the formulation in Lagrangian coordinates. Then, the equivalent strain is used as an independent space variable. In the case of elastic/plastic problems, this change of independent variables has proved advantageous for a class of problems [22–24].

A practical aspect of the solution is that simple solutions are essential for estimating the required forming pressure in tube hydroforming of monometallic and clad tubes [25,26]. Moreover, the solution can be used as a benchmark problem for verifying numerical codes, which is a necessary step before using such codes [27,28].

## 2. Statement of the Problem

Consider a two-layer tube of initial outer radius $R_b$ and inner radius $R_a$, subjected to uniform pressure $P$ over the inner radius. The outer radius of the inner layer and the inner radius of the outer layer is $R_c$ (Figure 1a). Each layer is rigid/plastic or hardening. The state of strain is plane. It is natural to adopt a cylindrical coordinate system $(r, \theta, z)$, the $z$-axis of which coincides with the tube's axis of symmetry. Then, the solution is independent of $\theta$. In particular, after any amount of deformation, the outer and inner radii of the tube are $b$ and $a$, respectively. The radius of the contact surface between the layers is $c$ (Figure 1b). Let $\sigma_r$, $\sigma_\theta$, and $\sigma_z$ be the normal stresses referred to the cylindrical coordinates. These stresses are the principal stresses. The circumferential velocity vanishes and the radial velocity is denoted as $u$.

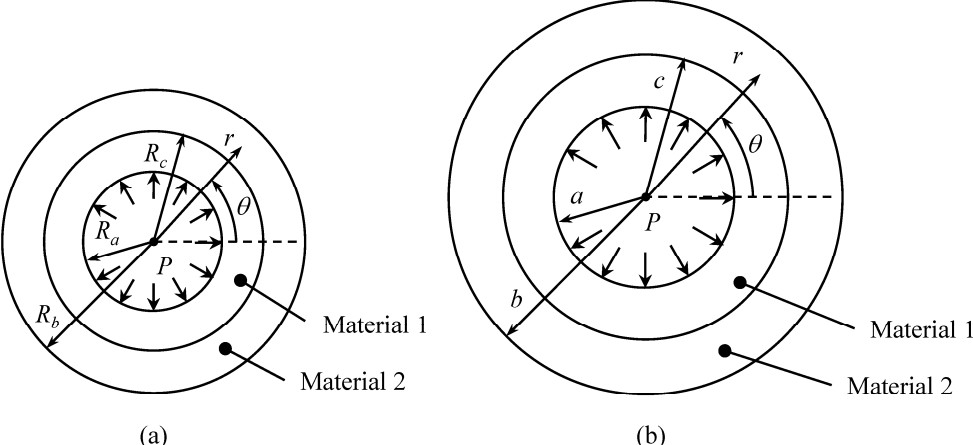

(a)                              (b)

**Figure 1.** Schematic diagram of the process: (**a**) initial configuration, (**b**) intermediate and final configurations.

In the case under consideration, any pressure-independent yield criterion reduces to

$$|\sigma_r - \sigma_\theta| = \chi \sigma_{eq}. \tag{1}$$

Here, $\sigma_{eq}$ is a measure of the equivalent stress and $\chi$ is constant. Let $\xi_r$ and $\xi_\theta$ be the non-zero normal strain rates referred to the cylindrical coordinates. The flow rule associated with the yield criterion (1) reduces to the equation of incompressibility

$$\xi_r + \xi_\theta = 0 \tag{2}$$

and the inequality

$$\xi_r(\sigma_r - \sigma_\theta) > 0. \tag{3}$$

It is assumed that $\sigma_{eq}$ is a function of the equivalent strain, $\varepsilon_{eq}$. The latter is determined from the equation:

$$\frac{d\varepsilon_{eq}}{dt} = \xi_{eq}. \tag{4}$$

Here, $t$ is the time, $d/dt$ denotes the convected derivative, and $\xi_{eq}$ is the equivalent strain rate. In the case under consideration,

$$\xi_{eq} = \chi|\xi_r| = \chi|\xi_\theta|. \tag{5}$$

Assuming that $\sigma_{eq}$ is equal to the axial stress in the uniaxial tension test, the von Mises yield criterion follows from Equation (1) at $\chi = 2/\sqrt{3}$ and Tresca's yield criterion at $\chi = 1$.

It is convenient to represent $\sigma_{eq}$ as $\sigma_{eq} = \sigma_0\Phi(\varepsilon_{eq})$, where $\sigma_0$ is the value of the equivalent stress at $\varepsilon_{eq} = 0$ and $\Phi(\varepsilon_{eq})$ is an arbitrary function of its argument, satisfying the conditions $\Phi = 1$ at $\varepsilon_{eq} = 0$ and $d\Phi/d\varepsilon_{eq} \geq 0$ for all $\varepsilon_{eq}$. Then, Equation (1) becomes

$$|\sigma_r - \sigma_\theta| = \chi\sigma_0\Phi(\varepsilon_{eq}). \tag{6}$$

The stress boundary condition is

$$\sigma_r = 0 \tag{7}$$

for $r = b$. The pressure $P$ is determined from the equation

$$P = -\sigma_r \tag{8}$$

where the radial stress is understood to be calculated at $r = a$.

### 3. General Solution

Since $\xi_r = \partial u/\partial r$ and $\xi_\theta = u/r$, Equation (2) can be immediately integrated to give

$$u = \frac{Ua}{r}. \tag{9}$$

Here, $U$ is the radial velocity at $r = a$. Since the material model is rate-independent, the magnitude of $U$ is immaterial. By definition, $\partial r/\partial t = u$. This equation and Equation (9) combine to give

$$\frac{\partial r}{\partial a} = \frac{a}{r}. \tag{10}$$

It has been taken into account here that $da/dt = U$. Equation (10) can be immediately integrated to give

$$r = \sqrt{R^2 + a^2 - R_a^2}. \tag{11}$$

Here, $R$ is the Lagrangian coordinate such that $r = R$ at $a = R_a$. One can solve Equation (11) for $R$ to obtain

$$R = \sqrt{r^2 - a^2 + R_a^2}. \tag{12}$$

It follows from Equation (10) that

$$\xi_r = \frac{\partial u}{\partial r} = -\frac{Ua}{r^2}. \tag{13}$$

One substitutes Equation (13) into Equation (5) to arrive at

$$\xi_{eq} = \chi \frac{Ua}{r^2}. \tag{14}$$

Equations (9) and (14) combine to give

$$\xi_{eq} = \chi \frac{Ua}{(R^2 + a^2 - R_a^2)}. \tag{15}$$

In Lagrangian coordinates, $d/dt = \partial/\partial t$. Therefore, Equation (15) becomes

$$\frac{\partial \varepsilon_{eq}}{\partial a} = \chi \frac{a}{(R^2 + a^2 - R_a^2)}. \tag{16}$$

It has been taken into account here that $da/dt = U$. One integrates Equation (16) to obtain

$$\varepsilon_{eq} = \frac{\chi}{2} \ln \left( \frac{R^2 + a^2 - R_a^2}{R^2} \right). \tag{17}$$

This solution satisfies the initial condition that $\varepsilon_{eq} = 0$ at $a = R_a$.

The only stress equilibrium equation that is not identically satisfied in the cylindrical coordinates is

$$\frac{\partial \sigma_r}{\partial r} + \frac{\sigma_r - \sigma_\theta}{r} = 0. \tag{18}$$

It is more convenient to rewrite this equation in the Lagrangian coordinates. It follows from Equations (11) and (12) that

$$\frac{\partial R}{\partial r} = \frac{r}{R}. \tag{19}$$

Then, Equation (18) becomes

$$\frac{\partial \sigma_r}{\partial R} \frac{\partial R}{\partial r} + \frac{\sigma_r - \sigma_\theta}{r} = \frac{\partial \sigma_r}{\partial R} \frac{r}{R} + \frac{\sigma_r - \sigma_\theta}{r} = 0. \tag{20}$$

Equations (3) and (13) show that $\sigma_r - \sigma_\theta < 0$. Then, using Equations (6) and (11), one transforms Equation (20) to

$$\frac{\partial \sigma_r}{\sigma_0 \partial R} = \frac{\chi R \Phi(\varepsilon_{eq})}{(R^2 + a^2 - R_a^2)}. \tag{21}$$

It is seen from the general structure of this equation that it is advantageous to use the equivalent strain as the independent variable instead of $R$. Using Equation (17), one replaces the differentiation with respect to $R$ in Equation (21) with the differentiation with respect to $\varepsilon_{eq}$ to attain

$$\frac{\partial \sigma_r}{\sigma_0 \partial \varepsilon_{eq}} = \frac{R^2 \Phi(\varepsilon_{eq})}{(R_a^2 - a^2)}. \tag{22}$$

Moreover, Equation (17) can be solved for $R$ to result in

$$R^2 = \frac{a^2 - R_a^2}{\exp(2\varepsilon_{eq}/\chi) - 1}. \tag{23}$$

Equations (20) and (23) combine to give

$$\frac{\partial \sigma_r}{\sigma_0 \partial \varepsilon_{eq}} = \frac{\Phi(\varepsilon_{eq})}{1 - \exp(2\varepsilon_{eq}/\chi)}.$$

(24)

The general solution of this equation is

$$\frac{\sigma_r}{\sigma_0} = \int\limits_{\varepsilon_0}^{\varepsilon_{eq}} \frac{\Phi(\omega)}{1 - \exp(2\omega/\chi)} d\omega + \frac{s_0}{\sigma_0}.$$

(25)

Here, $\omega$ is a dummy variable of integration, and $s_0$ is the value of $\sigma_r$ at $\varepsilon_{eq} = \varepsilon_0$. The circumferential stress is determined from Equations (6) and (25) as

$$\frac{\sigma_\theta}{\sigma_0} = \int\limits_{\varepsilon_0}^{\varepsilon_{eq}} \frac{\Phi(\omega)}{1 - \exp(2\omega/\chi)} d\omega + \chi \Phi(\varepsilon_{eq}) + \frac{s_0}{\sigma_0}.$$

(26)

Since $\varepsilon_{eq}$ is independent of $R$ at the initial instant, $\partial \varepsilon_{eq}/\partial R = 0$ at $a = R_a$. Therefore, the transformation of Equation (21) into Equation (22) is not justified at the initial instant. As a result, one cannot put $\varepsilon_0 = 0$ in Equation (25) and the following equations that involve this quantity. The solution at the initial instant is not required in the present paper. If one requires such a solution, then it is necessary to return to Equation (21). It is then necessary to put $\varepsilon_{eq} = 0$, which is equivalent to putting $\Phi(\varepsilon_{eq}) = 1$. The resulting equation can be immediately integrated in terms of elementary functions to provide the radial distribution of $\sigma_r$.

### 4. Expansion of a Two-Layer Tube

Throughout this paper's remainder, subscript *1* denotes quantities related to the inner layer and subscript *2* to the outer layer (Figure 1).

Let $\varepsilon_a$, $\varepsilon_b$, and $\varepsilon_c$ be the values of the equivalent strain at $R = R_a$, $R = R_b$, and $R = R_c$, respectively. Then, it follows from Equation (17) that

$$\varepsilon_a = \chi \ln\left(\frac{a}{R_a}\right), \quad \varepsilon_b = \frac{\chi}{2} \ln\left(\frac{R_b^2 + a^2 - R_a^2}{R_b^2}\right), \text{ and } \varepsilon_c = \frac{\chi}{2} \ln\left(\frac{R_c^2 + a^2 - R_a^2}{R_c^2}\right).$$

(27)

The solution to Equation (25) satisfies the boundary condition of Equation (7) if $s = s_0$ and $\varepsilon_0 = \varepsilon_b$. Then,

$$\frac{\sigma_r^{(2)}}{\sigma_0^{(2)}} = \int\limits_{\varepsilon_b}^{\varepsilon_{eq}} \frac{\Phi^{(2)}(\omega)}{1 - \exp(2\omega/\chi)} d\omega.$$

(28)

Let $\sigma_c$ be the value of the radial stress at $R = R_c$. It follows from Equation (28) that

$$\frac{\sigma_c}{\sigma_0^{(2)}} = \int\limits_{\varepsilon_b}^{\varepsilon_c} \frac{\Phi^{(2)}(\omega)}{1 - \exp(2\omega/\chi)} d\omega.$$

(29)

The radial stress must be continuous across the bi-material interface. Therefore, $\sigma_r^{(1)} = \sigma_c$ at $R = R_c$. The solution to Equation (25) satisfying this condition is

$$\frac{\sigma_r^{(1)}}{\sigma_0^{(2)}} = k \int\limits_{\varepsilon_c}^{\varepsilon_{eq}} \frac{\Phi^{(1)}(\omega)}{1 - \exp(2\omega/\chi)} d\omega + \frac{\sigma_c}{\sigma_0^{(2)}}$$

(30)

where $k = \sigma_0^{(1)}/\sigma_0^{(2)}$. Equations (29) and (30) combine to give

$$\frac{\sigma_r^{(1)}}{\sigma_0^{(2)}} = k\int\limits_{\varepsilon_c}^{\varepsilon_{eq}} \frac{\Phi^{(1)}(\omega)}{1 - \exp(2\omega/\chi)}d\omega + \int\limits_{\varepsilon_b}^{\varepsilon_c} \frac{\Phi^{(2)}(\omega)}{1 - \exp(2\omega/\chi)}d\omega \tag{31}$$

The pressure of the inner radius is determined from Equations (8) and (31) as

$$\frac{P}{\sigma_0^{(2)}} = -k\int\limits_{\varepsilon_c}^{\varepsilon_{eq}} \frac{\Phi^{(1)}(\omega)}{1 - \exp(2\omega/\chi)}d\omega - \int\limits_{\varepsilon_b}^{\varepsilon_c} \frac{\Phi^{(2)}(\omega)}{1 - \exp(2\omega/\chi)}d\omega \tag{32}$$

Together with Equations (17) and (26), the solution above supplies the dependence of the radial and circumferential stresses on the Lagrangian coordinate in parametric form, with the equivalent strain being the parameter. One can use Equation (12) to find the distribution of these stresses along the *r*-axis.

In general, the integral in Equation (25) should be evaluated numerically. However, two hardening laws of practical importance allow for the evaluation of this integral in terms of special functions. In the case of linear hardening, $\Phi(\varepsilon_{eq}) = 1 + \beta\varepsilon_{eq}$. Using this function, one finds

$$\int\limits_{\varepsilon_0}^{\varepsilon_{eq}} \frac{\Phi(\omega)}{1 - \exp(2\omega/\chi)}d\omega = \frac{\chi}{4}\left\{ \begin{array}{c} \beta\chi\mathrm{Li}_2\left[\exp\left(-\frac{2\varepsilon_{eq}}{\chi}\right)\right] - \beta\chi\mathrm{Li}_2\left[\exp\left(-\frac{2\varepsilon_0}{\chi}\right)\right] - \\ 2(1 + \beta\varepsilon_{eq})\ln\left[1 - \exp\left(-\frac{2\varepsilon_{eq}}{\chi}\right)\right] + 2(1 + \beta\varepsilon_0)\ln\left[1 - \exp\left(-\frac{2\varepsilon_0}{\chi}\right)\right] \end{array} \right\}. \tag{33}$$

Here, $\mathrm{Li}_2(\varepsilon_{eq})$ is the dilogarithm function. In the case of Voce's hardening law, $\Phi(\varepsilon_{eq}) = 1 + (\beta - 1)[1 - \exp(-n\varepsilon_{eq})]$. Using this function, one finds

$$\int\limits_{\varepsilon_0}^{\varepsilon_{eq}} \frac{\Phi(\omega)}{1 - \exp(2\omega/\chi)}d\omega = \beta(\varepsilon_{eq} - \varepsilon_0) - \frac{1}{2}\beta\chi\ln\left[\frac{1 - \exp(2\varepsilon_{eq}/\chi)}{1 - \exp(2\varepsilon_0/\chi)}\right] + \frac{(\beta - 1)}{n}\left[\exp(-n\varepsilon_{eq})\Lambda(\varepsilon_{eq}) - \exp(-n\varepsilon_0)\Lambda(\varepsilon_0)\right]. \tag{34}$$

Here, $\Lambda(x) \equiv 2F1[1, -n\chi/2, 1 - n\chi/2, exp(2\varepsilon eq/\chi)]$ is the hypergeometric function.

## 5. Experimental Verification and Illustrative Examples

Three geometric parameters for the initial configuration, the value of $\chi$, and two arbitrary functions classify the boundary value problem. Therefore, a detailed parametric analysis of the solution is not feasible. However, the solution is very simple for any given set of initial data. The numerical results below focus on the maximum value of the internal pressure, $P_{\max}$. Simple solutions for this quantity are essential for estimating the required forming pressure in tube hydroforming of monometallic and clad tubes [25,26]. The solution above certainly belongs to this class of solutions.

Experimental data on hydroforming of clad tubes are presented in [26]. The outer tube is made of aluminum alloy A1060-O and the inner tube from copper alloy C1020TS-O. The initial outer radius of the clad tube is 20 mm, and the initial total thickness of the two layers is 1.5 mm. The experiments have been carried out for several ratios of the outer tube's thickness to the inner tube's thickness. The stress–strain curves of the copper and aluminum alloys have been represented as

$$\sigma_{eq} = 518\varepsilon_{eq}^{0.45} \quad \text{and} \quad \sigma_{eq} = 144\varepsilon_{eq}^{0.25}, \tag{35}$$

respectively. Here, the equivalent stress is expressed in MPa. The equations in Equation (35) are not compatible with the restrictions imposed on the function $\Phi(\varepsilon_{eq})$ before Equation (6). Therefore, Equation (35) is replaced with Ludwik's hardening law

$$\sigma_{eq} = 31.3 + 496\varepsilon_{eq}^{0.507} \quad \text{and} \quad \sigma_{eq} = 32.2 + 115.7\varepsilon_{eq}^{0.37}. \tag{36}$$

Here, the first equation corresponds to the copper alloy and the second to the aluminum alloy. The difference between the laws in Equations (35) and (36) is negligible, and is revealed only at small strains (Figure 2). The solid curves correspond to Equation (35) and the broken curves to Equation (36). Using Equation (36) and assuming that $\chi = 2/\sqrt{3}$, one can find the function involved in Equation (6) as

$$\Phi^{(1)}\left(\varepsilon_{eq}\right) = 1 + 15.85\varepsilon_{eq}^{0.507} \quad \text{and} \quad \Phi^{(2)}\left(\varepsilon_{eq}\right) = 1 + 3.6\varepsilon_{eq}^{0.37}. \tag{37}$$

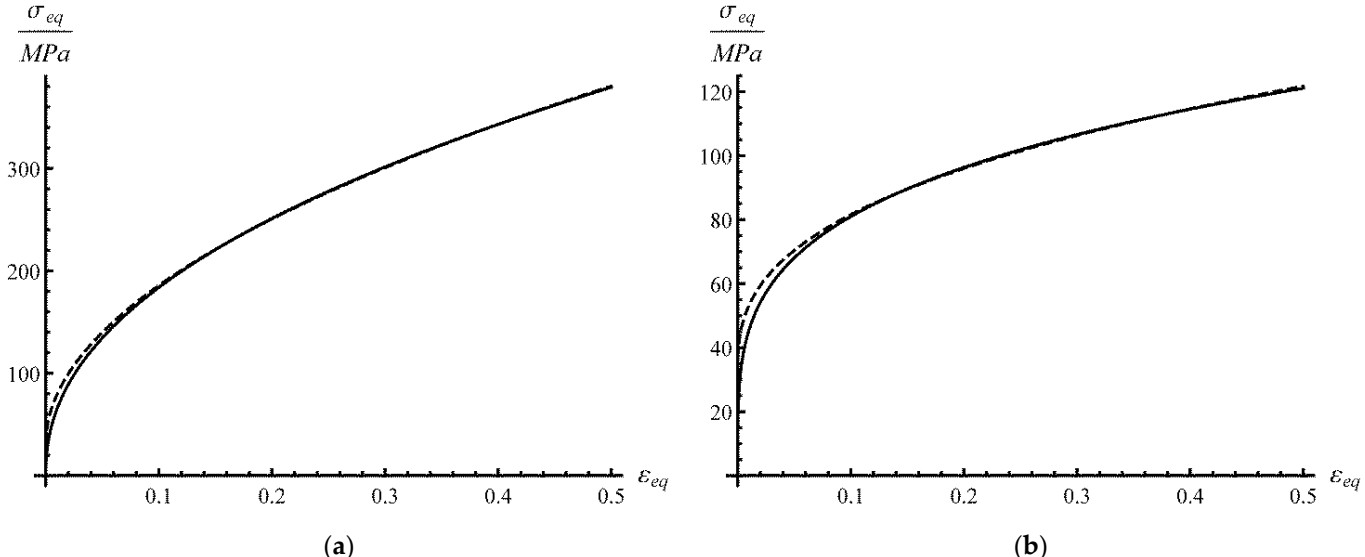

**Figure 2.** Stress–strain curves according to Equations (35) and (36): (**a**) copper alloy, (**b**) aluminum alloy. The solid curves correspond to Equation (35) and the broken curves to Equation (36).

Moreover, $\sigma_0^{(1)} = 31.3\,\text{MPa}$ and $\sigma_0^{(2)} = 32.2\,\text{MPa}$. The volume fraction of the copper alloy is determined as

$$\lambda = \left(\frac{R_c^2 - R_a^2}{R_b^2 - R_a^2}\right) \times 100\%. \tag{38}$$

Since $R_a$ and $R_b$ are fixed in [10], $\lambda$ is controlled by $R_c$. It is seen from the experimental data depicted in Figure 10 in [10] that $P_{\max}$ is practically a linear function of $\lambda$. This function can be interpolated as

$$P_{\max} = 6 + 9.5\left(\frac{\lambda}{100\%}\right). \tag{39}$$

Here, $P_{\max}$ is expressed in MPa. One can substitute Equation (37) into Equation (32) for calculating $P$ as a function of $a$. A local maximum of this function is found numerically. A comparison of the experimental data from [26] and the theoretical solution is shown in Figure 3. The solid line represents Equation (39) and the broken line is the theoretical solution found using Equation (32). It is seen from this figure that the theoretical solution is quite accurate.

As another example, a tube made of Al–Li alloy (Material 1 in Figure 1) and 5A06 aluminum alloy (Material 2 in Figure 1) is considered. Paper [20] provides the mechanical properties of these materials. In our nomenclature, $\sigma_0^{(1)} = 77.7\,\text{MPa}$ and $\sigma_0^{(2)} = 155\,\text{MPa}$. Moreover,

$$\Phi^{(1)} = 1 + 5.08\varepsilon_{eq}^{0.28} \quad \text{and} \quad \Phi^{(1)} = 1 + 4.4\varepsilon_{eq}^{0.3}. \tag{40}$$

The initial configuration is determined by

$$\frac{R_c}{R_a} = 1 + \frac{t_1}{R_a} \quad \text{and} \quad \frac{R_b}{R_a} = 1 + \frac{t_1 + t_2}{R_a}. \tag{41}$$

Here, $t_1$ is the initial thickness of the inner layer and $t_2$ is the initial thickness of the outer layer. In all calculations, $t_1 = 1.8\,\text{mm}$, $R_a = 100\,\text{mm}$, and $\chi = 2/\sqrt{3}$. Figure 4 depicts the variation of the inner pressure with the inner radius of the tube for three values of $t_2$. The value of $P$ increases with $t_2$. All three curves attain a local maximum at a certain value of $a$. The value of $P_{\max}$ can be found numerically with no difficulty.

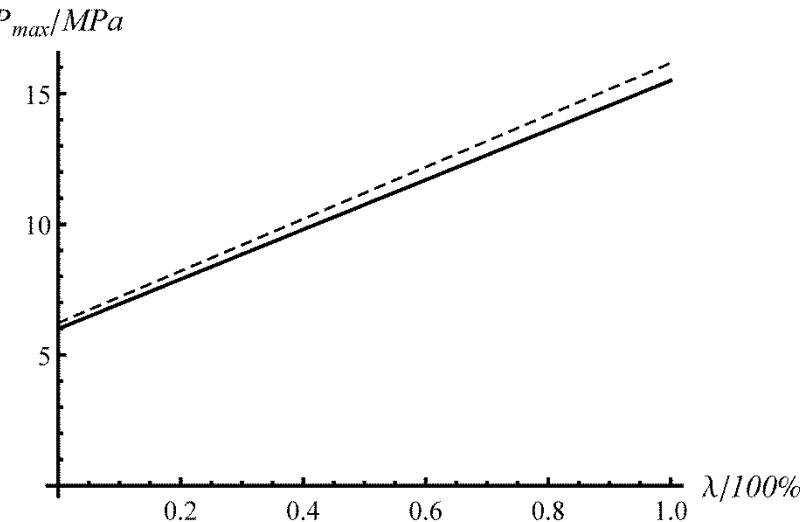

**Figure 3.** Comparison of the experimental data from [26], and the theoretical solution resulting from Equation (32). The solid line represents the experimental data and the broken line the theoretical solution found using Equation (32).

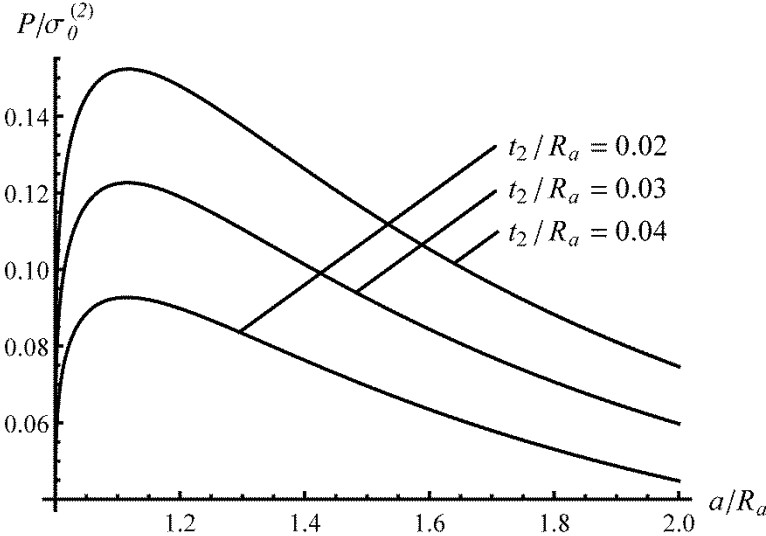

**Figure 4.** Dependence of the internal pressure on the inner radius for several initial thicknesses of the outer layer.

## 6. Conclusions

The solution presented describes the plane-strain expansion of a two-layer tube for tube hydroforming applications. The solution has been reduced to evaluating the integral in Equation (25). If the maximum value of the pressure is required, then a numerical technique is necessary. No restriction on the pressure-independent yield criterion and hardening law has been applied. Therefore, the solution is advantageous for the preliminary design of the tube hydroforming process. It is of special importance because many parameters and functions classify the boundary value problem. The solution's accuracy is verified by comparing it to the experimental data provided in [26] (Figure 3). As another example,

the effect of the thickness of the outer layer on the pressure applied over the inner radius of the tube has been investigated using the data provided in [20] (Figure 4). It is seen from this figure that its magnitude attains a local maximum at a certain stage of the process. It is probably because of competition between the change of geometric parameters and hardening. Simple solutions for P's maximum value are essential for estimating the required forming pressure in tube hydroforming of clad tubes [26].

The solution is exact. Therefore, it can be used for validating numerical solutions, which is a necessary step before their use in applications [27].

The solution in Section 3 is valid for any value of *a*. Therefore, it can be used to analyze and design hydroforging processes introduced in [29].

The solution in Section 4 is for two-layer tubes. The line of reasoning in this section shows that extending this solution to multi-layer tubes requires adding equations similar to Equation (31) for each interface, which is a relatively simple task.

**Author Contributions:** Formal analysis, S.A. and E.L.; supervision, L.L.; conceptualization, L.L.; writing—original draft, S.A. and E.L. All authors have read and agreed to the published version of the manuscript.

**Funding:** This research was made possible by the project AAAA-A20-120011690136-2 (Russian Ministry of Science and Education).

**Institutional Review Board Statement:** Not applicable.

**Informed Consent Statement:** Not applicable.

**Data Availability Statement:** Not applicable.

**Conflicts of Interest:** The authors declare no conflict of interest.

## Nomenclature

| | |
|---|---|
| $a$ | inner radius of the tube after any amount of deformation |
| $b$ | outer radius of the tube after any amount of deformation |
| $c$ | radius of the interface between the layers after any amount of deformation |
| $k$ | parameter introduced after Equation (30) |
| $P$ | pressure over the inner radius of the tube |
| $P_{max}$ | maximum inner pressure in the course of the hydroforming process |
| $R$ | Lagrangian coordinate |
| $R_a$ | Lagrangian coordinate of the inner radius of the tube |
| $R_b$ | Lagrangian coordinate of the outer radius of the tube |
| $R_c$ | Lagrangian coordinate of the interface between the layers |
| $(r, \theta, z)$ | cylindrical coordinate system |
| $t$ | Time |
| $t_1$ | initial thickness of the inner layer |
| $t_2$ | initial thickness of the outer layer |
| $U$ | radial velocity at the inner radius of the tube |
| $u$ | radial velocity |
| $\varepsilon_{eq}$ | equivalent strain |
| $\xi_{eq}$ | equivalent strain rate |
| $\xi_r$ and $\xi_\theta$ | radial and circumferential strain rates, respectively |
| $\sigma_{eq}$ | equivalent stress |
| $\sigma_r, \sigma_\theta$ and $\sigma_z$ | radial, circumferential, and axial stresses, respectively |
| $\sigma_0$ | initial yield stress in tension |
| $\phi$ | arbitrary function of the equivalent strain introduced in Equation (6) |
| $\chi$ | parameter introduced in Equation (1) |

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
