# Peer review of "Description of the Expansion of a Two-Layer Tube: An Analytic Plane-Strain Solution for Arbitrary Pressure-Independent Yield Criterion and Hardening Law"

_metals, doi:10.3390/met11050793_

Round 1

Reviewer 1 Report

In this work, authors proposed an analytical solution for describing the expansion the expansion of a two-layer tube under plane-strain conditions for its subsequent use for the preliminary design of hydroforming processes. The approach is of course interesting, but the manuscript lacks several scientific aspects to be considered for a publication at the current stage. Authors have not described the motivation properly in the introduction. They have not clearly discussed possible scientific insight one can gain by reading the manuscript. The referee could not get into the insight in the deformation of metal from this work, which is crucial for Metals. Authors must also illustrate the merit of this work. The only things gained by the referee from this paper is how authors have derived the analytical solution and a simple verification, which are way too less for the manuscript. By considering all of these aspects, the referee recommends to reject the manuscript in the current form. The authors must improve the quality of the manuscript.

Author Response

The corrections made in the revised manuscript are shown in red.

In this work, authors proposed an analytical solution for describing the expansion the expansion of a two-layer tube under plane-strain conditions for its subsequent use for the preliminary design of hydroforming processes. The approach is of course interesting, but the manuscript lacks several scientific aspects to be considered for a publication at the current stage. Authors have not described the motivation properly in the introduction. They have not clearly discussed possible scientific insight one can gain by reading the manuscript. The referee could not get into the insight in the deformation of metal from this work, which is crucial for Metals. Authors must also illustrate the merit of this work. The only things gained by the referee from this paper is how authors have derived the analytical solution and a simple verification, which are way too less for the manuscript. By considering all of these aspects, the referee recommends to reject the manuscript in the current form. The authors must improve the quality of the manuscript.

Answer.

The reviewer has correctly understood that the motivation of this work is to describe the expansion of a two-layer tube using an analytic solution. The only missed point here that the solution is valid for any pressure-independent yield criterion and hardening law. However, the title of the paper emphasizes the latter. Since the solution is valid for any pressure-independent yield criterion and hardening law, it is computationally much more efficient than any numerical solution, assuming that its experimental verification is provided. Experimental verification has been provided for Pmax (Fig. 3). Because this parameter is very important for the design of hydroforming processes, it already shows the manuscript's merit. It is evident from the general structure of the solution that it can be extended to multi-layer tubes with no conceptual difficulty.

We do not understand the comment “The referee could not get into the insight in the deformation of metal from this work, which is crucial for Metals”. Equations (11) and (12) connect the Lagrangian and Eulerian coordinates. These transformation equations contain the full information about any aspect of deformation.

It is our understanding that the reviewer has no critical comment related to the solution itself. Therefore, we have corrected the Introduction and Discussion.

Reviewer 2 Report

This paper reports innovative methods of calculating hydroforming, and should be published.  I have only very minor comments:

l. 59: plane, not plain

l. 99: Eq. 15, not 13

l. 199: Explain why the (normalized) internal pressure is a non-monotonic function of the thickness of the outer layer.

Author Response

We have incorporated all your comments in the revised manuscript. The corrections made in the revised manuscript are shown in red.

Reviewer 3 Report

This develops an analytical solution for the expansion of a two-layer tube under strain in a plane to enable initial designs of hydroforming processes. Hydroforming processes are important in metal forming and this paper presents a nice description of a simple model to use for the initial design of such processes.

The paper requires some improvement in the presentation as the overall work is quite nice but can be better presented.

The Introduction is a little rough. The English in the Introduction needs to be improved. They also need to provide an improved description of the hydroforming process and its importance in metal froamng to provide a better context of the work.

In Figures 2 and 3, label what the dashed and solid lines actually are.

The following may be a naïve question but do the authors have to worry about poles in equations 21 and 22? In other words, can the denominators ever go to zero?

Overall, the presentation and derivation of the equations is excellent as they give all of the steps.

It would be helpful to have a table with all of the variables defined and a summary of the experimental parameters that were used.

Author Response

The corrections made in the revised manuscript are shown in red.

  • The Introduction is a little rough. The English in the Introduction needs to be improved. They also need to provide an improved description of the hydroforming process and its importance in metal froamng to provide a better context of the work.

We have extended the Introduction. We hope that there are no obvious grammatic errors. We have checked it with the premium version of Grammarly. The style may not be so good. We are not native English speaking authors It is hard to expect perfect English from us.

  • In Figures 2 and 3, label what the dashed and solid lines actually are.

The meaning of the lines was described in the text of the original manuscript (lines 158, 159, 169, and 170). We have corrected the captions in the revised manuscript.

  • The following may be a naïve question but do the authors have to worry about poles in equations 21 and 22? In other words, can the denominators ever go to zero?

No, it is not a naïve question. It is our fault that we overlooked it. This issue does not affect the solution but should be explained. We have added an explanation concerning Eq.(22) after Eq.(26). The denominator in Eq.(21) never vanishes because it is equal to r^2 (please Eq.(12)).

  • It would be helpful to have a table with all of the variables defined and a summary of the experimental parameters that were used.

The table is provided in the revised manuscript. As to the experimental results, they are from the literature. We only know the information we refer to in our paper.

Round 2

Reviewer 1 Report

Authors have improved the introduction of the manuscript which is important for motivating potential reader. Before accepting the manuscript, the referee suggest author to work on the following point.

- In the typical publication, it is a must to have the chapter conclusion which is missing in the manuscript. After going through the manuscript, it would be better to change the chapter "Discussion" to "Conclusion". Furthermore, the last 2 sentences do not entirely match with the first paragraph of this section. The referee would then suggest authors to reformulate this section for a better flow.

Author Response

"In the typical publication, it is a must to have the chapter conclusion which is missing in the manuscript. After going through the manuscript, it would be better to change the chapter "Discussion" to "Conclusion". Furthermore, the last 2 sentences do not entirely match with the first paragraph of this section. The referee would then suggest authors to reformulate this section for a better flow."

We agree with the reviewer. Section 6 has been renamed. The last two sentences have been rewritten.